# The Role of Hand-Held Cardiac Ultrasound in Patients with COVID-19

**DOI:** 10.3390/biomedicines11020239

**Published:** 2023-01-17

**Authors:** Ziv Dadon, Shemy Carasso, Shmuel Gottlieb

**Affiliations:** 1Jesselson Integrated Heart Center, Shaare Zedek Medical Center, Jerusalem 9103102, Israel; 2The Azrieli Faculty of Medicine, Bar-Ilan University, Zefat 1311502, Israel; 3The Sackler Faculty of Medicine, Tel Aviv University, Tel Aviv 6997801, Israel

**Keywords:** COVID-19, echocardiography, feasibility studies, physiology: safety, ventricular function

## Abstract

The role of point-of-care ultrasound (POCUS) in patient management has been established in recent years as an important tool. It is increasingly used by multiple medical disciplines in numerous clinical settings, for different applications and diagnostic purposes and in the guidance of procedures. The introduction of small-sized and inexpensive hand-held ultrasound devices (HUDs) has addressed some of the POCUS-related challenges and has thus extended POCUS’ applicability. HUD utilization is even more relevant in the COVID-19 setting given the operators’ infection risk, excessive workload concerns and general equipment contamination. This review focuses on the available technology, usefulness, feasibility and clinical applications of HUD for echocardiogram assessment in patients with COVID-19.

## 1. Introduction

Cardiovascular disease is a well-established risk factor for morbidity and mortality among hospitalized coronavirus disease 2019 (COVID-19) patients, with worse outcomes among COVID-19 patients with abnormal echocardiography results [1,2]. Furthermore, it has been demonstrated that elevation in cardiac biomarkers, e.g., high-sensitivity cardiac troponin-I, is associated with a poor prognosis in COVID-19 patients [3]. Accordingly, echocardiography is an important tool in the clinical management of patients hospitalized with COVID-19 [4]. However, the routine echocardiographic assessment of COVID-19 patients is currently discouraged due to the risk of infection of echocardiography professional operators, concerns of excessive workload and general equipment contamination in the setting of the pandemic [5,6,7]. In this setting, hand-held ultrasound device (HUD) can be utilized as an instantaneous clinical diagnostic tool that minimizes clinician–patient interaction. This review focuses on the available technology, usefulness, feasibility and clinical applications of HUD for echocardiography in patients hospitalized with COVID-19.

## 2. The Rational of HUD-Based Cardiac POCUS Utilization in COVID-19 Setting

Point-of-care ultrasound (POCUS) in the practice of different medical disciplines continues to evolve with a recognition of its role as a valuable adjunct to the delivery of excellent clinical care [8,9]. POCUS is utilized for many clinical applications, including resuscitative, diagnostic, procedural, therapeutic and monitoring [10]. POCUS has numerous benefits when compared to traditional ultrasound devices, including shortened time-to-diagnosis and treatment, reduction in use of radiation-requiring imaging studies and procedures, reduced cost, portability, dynamic and focused exam, bedside evaluations and high availability, including for follow-up examinations [11,12]. These advantages may even be more marked when it comes to HUD use [13]. These benefits have the potential to play a significant role in COVID-19 milieu, as its special setting requires evaluation, diagnosis and treatment in a strictly contained environment. The miniature and inexpensive HUD platform used for bedside POCUS echocardiography would provide a real-time instantaneous evaluation, and thus may have an immediate impact on clinical management. Moreover, these devices are wireless, internal battery-operated, mostly screen-touched that could be assigned to specific departments. As they are free from unnecessary trays, baskets, transducer holders’ brackets, power and data wires, their utilization may dramatically reduce the risk for cross-infection transmission [14,15,16]. As compared with conventional devices, HUDs also enable real-time Doppler (2D and color) as well as field-of-view (2D and color flow) with similar maximum depth (for example, up to 30 cm for the single probe of Butterfly iQ™, Butterfly Network Inc., Guilford, CT, and up to 24 cm for both the sector array transducer of Lumify™, Philips, Eindhoven, the Netherlands, and the phased array transducer of Vscan Extend™, GE Healthcare, Chicago, IL), automatic frame rate, and transducer frequency range (for example, 1–10 MHz for Butterfly iQ™, 1–4 MHZ for Lumify™ and 1.7–3.8 MHz for Vscan Extend™) with image quality adequate for interpretation and a correct diagnosis in most cases [17]. Nonetheless, HUD use may involve several challenges, including small display size, short-lasting battery, limited imaging quality, lack of advanced measurements and unclear findings requiring confirmation by a high-end device [17]. Advantages, limitations and similarities of conventional vs. HUD use for echocardiographic evaluation in COVID-19 setting are summarized in Table 1.

## 3. HUDs: Probe Technology and Equipment

To allow cost reduction with fewer complexities, technologic innovations have been incorporated, replacing the traditional use of piezoelectric crystals with a novel ultrasound-on-chip technology method that incorporates a two-dimensional array of 9000 micro sensors capable of emulating different transducers over a small frequency range [18]. This technology bypasses the traditional connection necessities between the probe and a specific designated display, enabling the image to be displayed on a nonspecific mobile device, thus reducing costs and further increasing portability.

The range of HUDs have recently been introduced into practice with differing physical characteristics, probe availability, technologies, connectivity and applications. The ultrasound device transducer is the most important component of the imaging system, and the quality of the images mainly relies on its performance. Accordingly, many of the HUDs are in fact hand-held ultrasound probes that can be connected to a wide array of commercially available mobile devices. Table 2 summarizes examples of different HUDs and probes, including type and manufacturer, relevant cardiac probe, physical aspects, display platforms (designated partnered console vs. commercially available mobile devices) and wireless function capabilities. When it comes to choosing a device, among the considerations to be included are the following: dedicated display platform vs. mobile tablet support, cabled probe vs. cableless, dual probe (two-headed transducer), durability, prominent buttons, weight, battery life, clips quality, depth, operating band width to support multiple imaging modes, artificial intelligence-based tools, touch screen, company support availability and advanced features (3D, Doppler types, captured temporal features, etc.).

## 4. POCUS Usefulness for Echocardiography Assessment in COVID-19 Patients

There are growing data supporting the utilization of POCUS for the diagnosis, evaluation and management of COVID-19 patients. Lung POCUS has a good diagnostic performance in a COVID-19 setting and can be used for COVID-19 diagnosis at patient’s home for identifying respiratory complications as well as a screening tool among symptomatic patients [19]. POCUS can also be incorporated as an important tool for the cardiovascular system assessment with different available scanning protocols [20]. The American Society of Echocardiography states that POCUS examinations performed by clinicians who are already caring for the patients present an attractive option for echocardiographic assessment by reducing unnecessary clinician–patient exposure [6].

The long-lasting debate as to whether diagnostic ultrasonography is superior to cardiac physical examination is even more relevant in COVID-19 clinical assessment [21]. Patient management in a COVID-19 setting involves personal protective equipment and strict infection control, measures that significantly limit the ability to perform the traditional physical examination, especially when it comes to stethoscope use [22]. It has been shown that POCUS has a higher diagnostic performance in patients with dyspnea than physical examination and stethoscopes in heart failure and pneumonia diagnosis [23]. In addition, POCUS use enables the diagnosis of cardiac entities that do not manifest as clearly as other diseases when addressed by physical examination (e.g., cardiac tamponade, pneumothorax, pulmonary embolism) [24].

Studies conducted on the POCUS-based management of patients hospitalized with COVID-19 have found a relatively high rate of cardiac abnormalities ranging up to 68%, with right ventricular (RV) systolic dysfunction being the most predominant finding (10–52.8%) [1,25,26,27,28,29,30]. A cross-sectional study demonstrated an association between disease severity and the prevalence of abnormal echocardiography results when comparing patients with non-severe COVID-19 to those with severe disease [31]. The study demonstrated larger biventricular diameters alongside with lower left ventricular ejection fraction (LVEF) and RV fractional area change (FAC) in those with severe disease. Furthermore, an association between cardiac abnormalities on echocardiography and mortality was found for both left ventricular (LV) longitudinal strain and RV free-wall strain. These results are less applicable for routine use as strain measurement is not generally performed in POCUS-based exams [32]. Lastly, the abnormal echocardiographic findings in patients admitted with COVID-19 can lead to management change in nearly one-third of tested patients [30]. These data suggest that the use of cardiac POCUS among other imaging modalities has the potential for clinical management adjustment and prognostic value among COVID-19 patients while minimizing the risk of contamination.

## 5. HUD Feasibility and Quality for Echocardiographic Assessment in COVID-19 Patients

Cardiac POCUS using a HUD has been widely adopted as a routine diagnostic tool in critical care as well as in other settings. However, the clinical data regarding HUD use for echocardiography assessment in COVID-19 setting is limited. The majority of COVID-19 POCUS-based studies were performed with standard ultrasound machines that are full-size and mobile.

A retrospective controlled study from Connecticut investigated the feasibility of HUD use (Lumify™) for cardiac evaluation among COVID-19 patients [33]. They showed that all HUD-based examinations (n = 90) were deemed to be diagnostic and provided sufficient information for the clinical care team. Despite the high rate of intubated patients (n = 57; 63.3%), no repeated echocardiography studies were required due to the inadequate imaging of the preliminary study performed using HUD. Similarly, in a recent prospective study conducted among 103 consecutive COVID-19 patients hospitalized in designated medical wards, we demonstrated through a blinded, fellowship-trained echocardiographer that only 13% of HUD-based studies (Vscan Extend™ with the dual probe) were categorized as having poor quality (Figure 1) [34].

The quality of RV demonstration was also high, with good/fair quality in 91% of the exams. The real-time interpretative accuracy of HUD-based LVEF assessment by the operators during the exam acquisition was reliable with fair to good correlation (r = 0.679, *p* < 0.001) and substantial agreement (Kappa = 0.612, *p* < 0.001) between the operator and the gold standard (Figure 2) [34].

LVEF agreement was also assessed using the Bland–Altman analysis revealing a mean bias of −0.96 (95% limits of agreement 9.43 to −11.35; *p* = 0.075). However, with regard to RV systolic function, only a fair agreement was demonstrated as compared with expert echocardiographer (Kappa = 0.308, *p* = 0.002).

In accordance with current COVID-19 guidelines recommending a shortened sonography time, the scan time using a HUD ranged from a mean of 5 ± 2 to 9 ± 3 min, a reduction of up to 79% in scanning time as compared with the conventional device controlled scan (a mean of 24 ± 7 min) [33,34]. Similarly, Maheshwarappa et al. found, in a prospective controlled observational study, a significant reduction of 55% in the scanning time using HUD device as compared with a standard device from a median of 20 (IQR of 17–22) minutes down to 9 (IQR of 8–11) minutes; *p* < 0.001 [35]. Furthermore, the total time duration spent in the patient’s room decreased by 71% [33].

In addition, HUD battery usage was good (14 ± 5% of battery capacity) with reasonable operator-to-patient proximity (59 ± 11 cm) leading to good perceptual measurements of operator safety [34].

Additionally, following the manufacturer-recommended protocol, the time required for disinfection decreased by 86% using a HUD as compared with conventional device [33].

## 6. The Usefulness and Diagnostic Yield of HUD for Echocardiography Assessment in COVID-19 Patients

The usefulness of HUD for echocardiographic assessment in non-COVID-19 settings was shown to be accurate for LV systolic function evaluation as compared with formal echocardiography [36,37], though concerns were raised regarding its use for valvular abnormalities assessment in case of moderate/severe pathologies or in case of valvular stenosis [38,39]. However, data regarding the usefulness of HUD for cardiac assessment in COVID-19 setting is more limited. Maheshwarappa et al. found that HUD-based cardiac assessment in COVID-19 patients was accurate and reliable when compared with conventional device assessment [35]. In a recent prospective study, we have demonstrated that COVID-19 hospitalized patients with an abnormal HUD-based echocardiogram were older and more likely to suffer from comorbidities and the use of chronic heart failure medications [40]. Additionally, HUD-based abnormal echocardiogram (defined as abnormal ventricular function/size or significant valvular pathology) was associated with worse endpoints (Figure 3). 

Abnormal echocardiogram independently predicted the composite endpoint (OR 6.19; 95% CI 1.50–25.57, *p*= 0.012). Another important finding was that among low-risk patients (room-air oxygen saturation ≥ 94%), the prevalence of the composite endpoint was very low (3.1%) with a low positive predictive value for HUD use in this group of patients. These results indicate that the utilization of a HUD is an important “rule-out” tool among COVID-19 high-risk patients and should be integrated early into their routine evaluation. HUD can be utilized for different echocardiographic clinical applications in the setting of COVID-19 infection. Figure 4 shows examples of routine HUD-acquired echocardiographic images in COVID-19 patients. 

There are several case reports/series describing HUD utilization in acute-care settings among COVID-19 patients for wide range of diagnoses [41,42,43,44]. A comprehensive list of HUD-based echocardiographic assessment in a COVID-19 setting, including evaluated entities, parameters and potential diagnoses, is detailed in Table 3. 

Different protocols were proposed for cardiac POCUS [20]. HUD-based echocardiography has a significant role in the diagnosis of COVID-19 associated abnormal processes, including hyperdynamic cardiac function, stress-induced cardiomyopathy, RV enlargement, pericardial effusion, acute pulmonary hypertension (secondary to pulmonary embolism or to the detrimental effects on the lung parenchyma) and diffuse myocardial inhibition, among other less common diagnoses. Figure 5 shows examples of abnormal findings using HUD for echocardiography real-time assessment among COVID-19 patients.

## 7. Evolving Technology and Future Perspectives

Newer designs of miniaturized HUD and wireless transducers are being evaluated, including wearable, belt-like and vascular imaging [45]. Tissue Doppler Imaging (TDI), three-dimensional (3D) and advanced technologies are mostly lacking or limited in currently available devices. With the further development of the novel ultrasound-on-chip technology, more applications would be readily available for real-time use. Additionally, the increasing repertoire of available artificial intelligence applications is expected to advance even more in coming years, including automatic indices measurements, quality grading and acquisition real-time improvement feedback [46,47]. These tools can be used as decision support tools for diagnostic accuracy improvement, acquisition optimization, multiple image interpretation integration and reduction in variability. Many of these applications are mainly relevant for non-expert operators. Moreover, the combination of image digitalization and tele-robotics may expand the use of ultrasound even further, enabling experts to perform an examination from a distance, thus virtualizing both interpretation and ultrasound image acquisition via semi-autonomous robotic system [38].

## 8. Conclusions

HUD-based echocardiographic assessment is a useful and accurate adjunctive tool in the management of COVID-19 patients. The portability, low-cost and instantaneous nature of this easily infection-controlled imaging modality is invaluable for the pulmonary and cardiovascular real-time assessment in COVID-19 settings. HUD utilization also involves the shortening of operator-patient exposure time and can be incorporated early into routine patient management as a diagnosis support tool for risk stratification and clinical management tailoring. Offering department-specific designated devices and carrying no risk of radiation exposure, HUD can be utilized to augment the limited physical examination in heterogenous COVID-19 patient population, including pediatrics and pregnant patients. However, HUD is less suitable for advanced hemodynamic valvular assessment. As such, patients with valvular lesions are suspected to be beyond mild in severity and cases with equivocal findings should be referred for a formal echocardiogram using a high-end device. Additionally, the routine utilization of focused echocardiogram among COVID-19 low-risk patients is not recommended for prognostication or as a screening tool. Further research and new modalities are required to further establish the role of HUD in a COVID-19 setting.

## Figures and Tables

**Figure 1 biomedicines-11-00239-f001:**
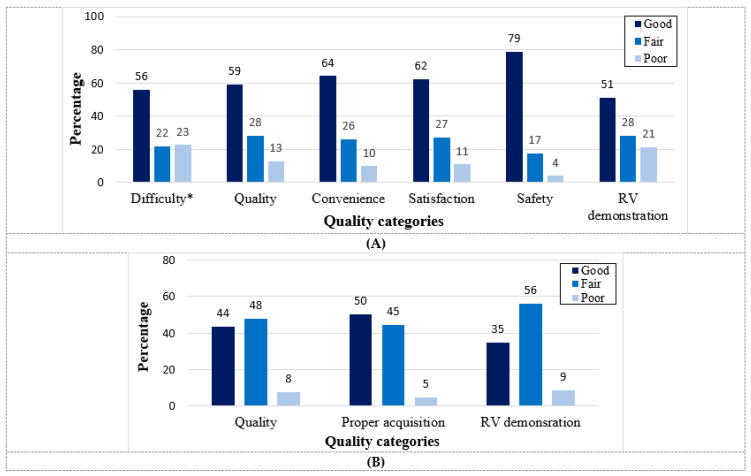
Quality, utilization and safety indices of hand-held echocardiogram according to operator and echocardiographer. Parameters were graded into three groups: Good, Fair and Poor. (**A**): Technical variables of hand-held echocardiogram included the difficulty, general quality, convenience, satisfaction, safety and demonstration of the RV, as documented manually by the operator at the patient bedside. * The gradings for the difficulty category were the following: Not difficult, Fairly difficult and Very difficult. (**B**): Technical variables included general quality, proper acquisition and RV demonstration, as documented by the echocardiographer during offline evaluation. Abbreviation: RV, right ventricle. Reproduced from Dadon et al [34]. with permission from Wiley.

**Figure 2 biomedicines-11-00239-f002:**
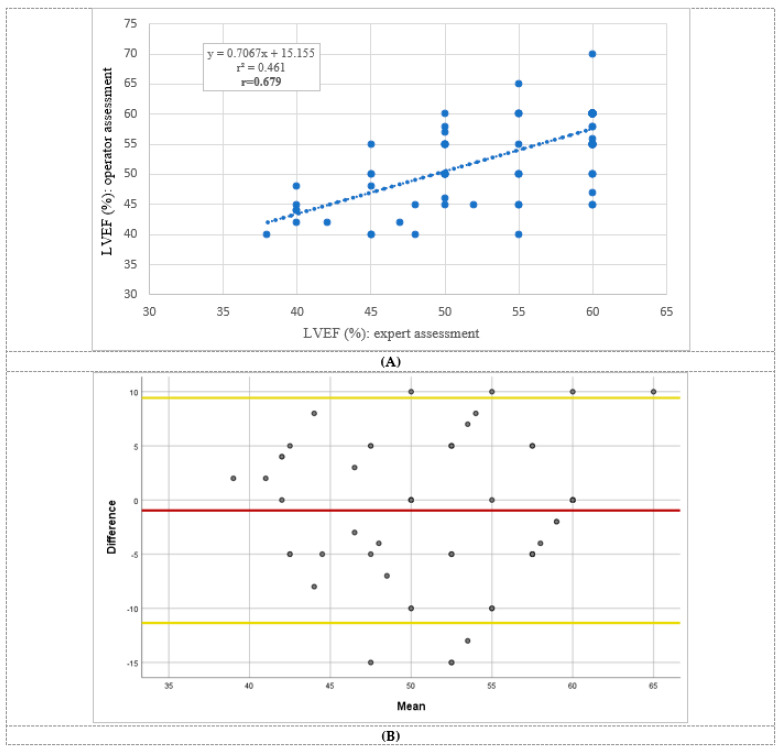
Left ventricular ejection fraction (LVEF) assessment correlation and agreement between the operator and the echocardiographer. (**A)**: LVEF correlation between the operator and the echocardiographer; Pearson correlation coefficient = 0.679 (*p* < 0.001). (**B**): LVEF assessment agreement using the Bland–Altman analysis revealed a mean bias of −0.96 as represented by the red line with limits of agreement ranging from 9.43 to −11.35 as represented by the yellow lines (*p* = 0.075). Abbreviations: LVEF, left ventricular ejection fraction. Reproduced from Dadon et al [34]. with permission from Wiley.

**Figure 3 biomedicines-11-00239-f003:**
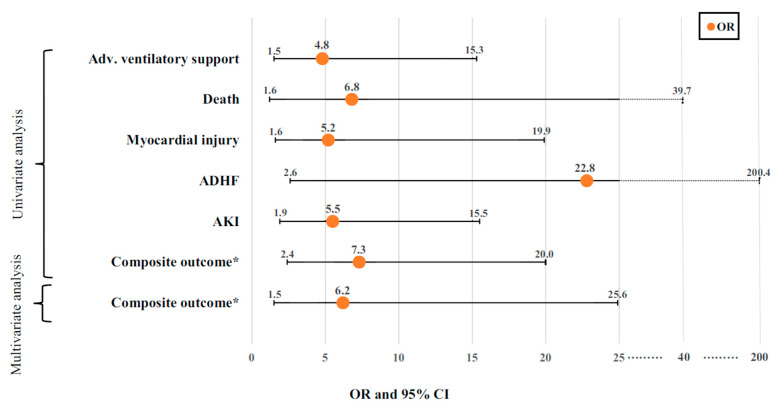
Significant associations (odds ratio and 95% confidence interval) between abnormal echocardiogram and serious adverse events (endpoints) * The primary endpoint was defined as a composite endpoint of in-hospital death, mechanical ventilation, shock and acute decompensated heart failure. Abbreviations: ADHF, acute decompensated heart failure; Adv., advanced; AKI, acute kidney injury; OR, odds ratio. Reprinted from the *Canadian Journal of Cardiology*, Volume 38, Issue 3, Dadon et al., The Utility of Handheld Cardiac and Lung Ultrasound in Predicting Outcomes of Hospitalised Patients With COVID-19, Pages 338–346, 2022, [40] with permission from Elsevier.

**Figure 4 biomedicines-11-00239-f004:**
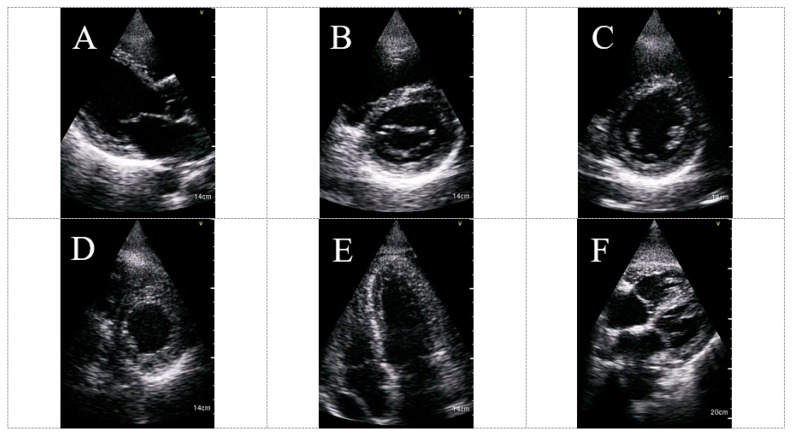
HUD acquired echocardiographic images in COVID-19 patients: parasternal long axis view (**A**), parasternal short axis view (MV level) (**B**), parasternal short axis view (PM level) (**C**), parasternal short axis view (apex level) (**D**), apical four chambers view (**E**) and subcostal view (**F**). The imaged were acquired using the Vscan Extend with the Dual Probe, General Electric. Abbreviations: HUD, hand-held ultrasound device; MV, mitral valve; PM, papillary muscles.

**Figure 5 biomedicines-11-00239-f005:**
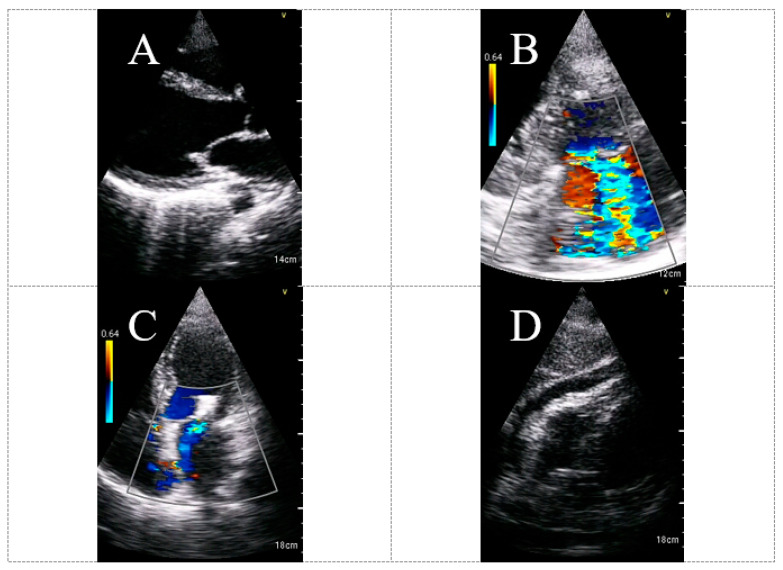
HUD acquired echocardiographic images in COVID-19 patients demonstrating abnormal findings: Dilated LV with reduced systolic function (**A**), severe MR (**B**), status post MitraClip transcatheter mitral valve repair (**C**) and pericardial effusion (subcostal view) (**D**). The imaged were acquired using the Vscan Extend with the Dual Probe, General Electric. Abbreviations: HUD, hand-held ultrasound device; LV, left ventricle; MR, mitral regurgitation.

**Table 1 biomedicines-11-00239-t001:** Advantaged and challenges * in HUD utilization for echocardiographic assessment in COVID-19 setting as compared with conventional devices.

Advantages	Challenges	Similarities ^†^
Very simple to use	Reduced spatial and temporal resolution	2D and color Doppler are in real-time
Portability (light-weighted, operated on battery, with potential wireless probes)	Small screen size with limited number of basic functions with no advanced modalities	Field-of-view (2D and color flow)
Bedside evaluation with reduced time to performance and cost effectiveness	Evaluation is mostly semi quantitative or qualitative (a bimodal assessment)	Maximum depth (see text)
Reduced cost	Doppler wave measurement is not available in all devices	Automatic frame rate
Easily infection control (minimized cavities/buttons with easy sterile wrapping and disinfection)	Simultaneous ECG, TDI, 3D and advanced technologies are mostly lacking	Transducer frequency (see text)
Availability for early diagnosis and follow-up examination	Privacy and security concerns	Correct final diagnosis in most cases

* Generalization was required in order to compare between the two device classes; thus, some of the mentioned differences may less apply to specific HUD devices. ^†^ Discrepancies may exist with regard to indices quality and max measurements. Abbreviations: cm, centimeter; ECG, ultrasound device; TDI, tissue Doppler imaging.

**Table 2 biomedicines-11-00239-t002:** Examples of different type and manufacturer of cardiac HUD.

Type of Device *	HUD Photo ^†^	Manufacturer/Headquarters	Cardiac Probe	Size (mm) **	Weight (gr.)	Display Platform (Screen Size)	Wireless Probe
BenQ H1300	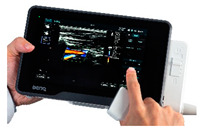	BenQ/Taipei City, Taiwan	2–4 MHz phased	S: 356 × 242 × 153	S: 1200	Designated console (8”)	-
Butterfly iQ+	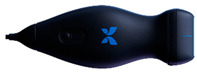	Butterfly Network/Burlington, Massachusetts	2–10 MHz phased	P: 163 × 56 × 35	P: 309	Mobile device	-
Chison SonoEye	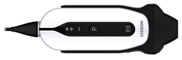	Chison/Wuxi, Jiangsu, China	2.5 MHz phased	P: 173 × 64 × 24	P: 100	Mobile device	-
Clarius PA HD3	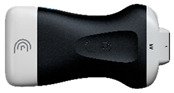	Clarius/Vancouver, BC, CA	1–5 MHz phased	P: 148 × 76 × 32	P: 292	Mobile device	√
Kosmos Ultraportable Ultrasound	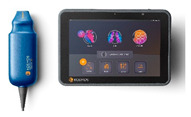	EchoNous/Redmond, WA	2.0–5.0 MHz phased	S: 216 × 146 × 59 P: 150 × 56 × 35	S: 653 P: 225	Designated console (8”) or Android tablet	-
LeSONO LU710PA	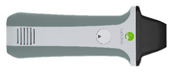	LELTEK Inc./Taipei, Taiwan	1.7–3.7 MHz phased	P: 194 × 74 × 40	P: 350	Mobile device	√
Lumify S4-1	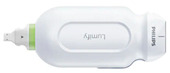	Philips Medical Systems/Eindhoven, Holland	1–4 MHz sector/phased	P: 10.2 × 5.1	P: 96	Mobile device	-
Vave health	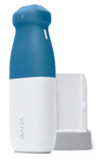	Vave Health/Santa Clara, California	1.5–3.5 MHz phased	P: 169 × 54 × 38	P: 340 (w)	Mobile device	√
Vscan Extend	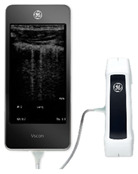	GE Healthcare/Chicago, Illinois	1.7–3.8 MHz Sector	S: 168 × 76 × 22 P: 129 × 39 × 28	S: 321 P: 85	Designated console (5.7”)	-

* Devices are arranged in an alphabetical order. A representative wired and wireless devices were chosen from each company. Technical details were taken from formal websites. The probes weight is with battery (whenever available). ^†^ Photos published with permission from the manufacturers. All rights reserved. ** Screen and probe size include the following measures: width × height × depth and length × width × depth, respectively. Abbreviations: HUD, hand-held ultrasound device; mm, millimeter; P, probe; S, screen.

**Table 3 biomedicines-11-00239-t003:** Characteristics of echocardiographic findings using HUD in different COVID-19 clinical settings of myocardial involvement.

Assessed Entity/Chamber	Evaluated Parameters and Findings	Potential Diagnoses
LV ^○^	Semi-quantitative and qualitative LV ejection fraction LV systolic dysfunction: Wall motion abnormalities vs. global dysfunction Cardiomyopathies pattern (dilation and hypertrophy)	Myocarditis Takotsubo cardiomyopathy Myocardial infarction
Pericardial space ^○^	Effusion: distribution and size (small: <10 mm; moderate: 10–20 mm; or large: >20 mm) Evidence for diastolic collapse	Pericarditis Tamponade
RV	RV size and function RV thrombus McConnell’s sign *	Acute pulmonary hypertension Pulmonary embolism
IVC	Diameter Inspiratory collapse	Part of volume status/shock assessment ^†^
Valvular function ^○^	Gross valvular morphology and mobility Color Doppler transvalvular flow qualitative assessment	Regurgitation Stenosis
Others	Cardiac masses Extra cardiac findings	Thrombus/tumor Pleural effusion Lung congestion

^○^ Some of the HUDs can measure the pulsed and wave Doppler and can potentially be used for the assessment of diastolic dysfunction, tamponade reciprocal respiration-related variations and valvular semi-quantitative and quantitative assessment; however, they required prior specific training and knowledge and are less readily available for non-expert operators in COVID-19 settings. * RV free wall akinesis with the sparing of the apex (in patients with RV strain, the sign can be useful to differentiate between pulmonary hypertension and pulmonary embolism). ^†^ Shock assessment using HUD echocardiography can be based on LV functionality and size, LV ejection fraction, IVC size and collapsibility, pericardial effusion and evidence for pulmonary congestion. Abbreviations: HUD, hand-held ultrasound device; IVC, inferior vena cava; LV, left ventricle; mm, millimeter; RV, right ventricle.

## Data Availability

No new data were created or analyzed in this study. Data sharing is not applicable to this article.

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
