# Peer review of "The Role of Hand-Held Cardiac Ultrasound in Patients with COVID-19"

_biomedicines, 2023, doi:10.3390/biomedicines11020239_

Round 1

Reviewer 1 Report

Dadon and collages present a review that highlight the relevance of miniature and low-cost hand-held ultrasound devices (HUDs) use in the COVID19 scenery.

The topic is relevant, the information in the context of the pandemic is useful and the manuscript is well documented.

Author Response

Thank you so much for your kind review.

Shmuel Gottlieb, MD

Reviewer 2 Report

The authors presented an interesting and excellent review of hand-held ultrasound devices (HUDs) to address some clinical needs in the pandemic setting. The point-of-care ultrasound has its own clinical significance even without the COVID-19 challenge, but this review may shed more lights on the usefulness of POCUS for the clinical practices with several necessary performances for reliable diagnosis. The overall manuscript is well written and organized, but I have some suggestions for the authors to further improve the impact of this work.

1. The comparisons of specs and performances among all the available HUDs would be useful, and the readers may want to know the authors opinion on choosing specific HUD for practical need.

2. Butterfly iQ and iQ+ may have some unique features, such as the wide operating band to support multiple imaging modes. The authors may also consider the comparisons on the temporal features captured by the HUDs.

3. In line 42, it is quite redundant to read “, availability and the high availability for follow-up examinations”. Please rewrite this sentence.

4. Panel A and Panel B in Figure 1 and Figure 2 are not appropriately labeled.

Author Response

We thank the reviewer for their helpful comments that enable us to amend the manuscript.

  1. Table 2 was edited with updated photos of the different available devices as well as the headquarters of the manufacturers. As we contacted the manufacturers for verification of the details and permission to publish the photo and especially as we have no previous experience with most of these devices, we feel that it'll be unfitting for us to express our thoughts recommending a specific device.

In accordance with the reviewer's suggestion, we added a short paragraph focusing on choosing a specific HUD for practical needs: " When it comes to choosing a device, among the considerations include a dedicated display platform vs mobile tablet support, cabled probe vs cableless, dual probe (two-headed transducer), durability, prominent buttons, weight, battery life, clips quality, depth, operating bandwidth to support multiple imaging modes, artificial intelligence-based tools, touch screen, company support availability, and advanced features (3D, Doppler types, captured temporal features, etc.)”.

  1. These special HUD’s features were added to the new paragraph quoted in Answer 1. Also, to focus on cardiac-designated devices, the table was edited, and non-relevant devices were removed. The headquarters location as well as a representative photo (with permission) were added.
  2. Corrected: "and the high availability, including for follow‐up examinations."
  3. As Figures 1 and 2 were reproduced from previous publications, they were designed in a similar fashion.

Shmuel Gottlieb, MD